# Reflection on Guangzhou's Strategic Spatial Planning: Current Status, Conflicts, and Dilemmas

**Miaoxi Zhao** [1,2,*], **Yuexi Yao** [1] **and Galuh Syahbana Indraprahasta** [3,4]

1   School of Architecture, South China University of Technology, Guangzhou 510640, China;
    aryuexi@mail.scut.edu.cn
2   State Key Laboratory of Subtropical Building and Urban Science, South China University of Technology,
    Guangzhou 510640, China
3   Center for Regional System Analysis, Planning and Development, IPB University, Bogor 16820, Indonesia;
    galuh.syahbana.indraprahasta@brin.go.id
4   Research Center for Population, National Research and Innovation Agency Indonesia (BRIN),
    Jakarta 12710, Indonesia
*   Correspondence: arzhao@scut.edu.cn; Tel.: +86-138-2602-5180

**Abstract:** Strategic spatial planning plays a pivotal role in effectively providing solutions for urban issues. In 2000, Guangzhou took the pioneering step of formulating China's first strategic spatial plan, known as the Guangzhou Strategic Plan. However, existing research has predominantly focused on the content of the "Guangzhou Strategic Spatial Plan" while lacking attention to its implementation effects. To address this empirical gap, this paper analyzes Guangzhou's current spatial structure from the perspectives of facilities, population, and industry. The results reveal that: (1) the effectiveness of the "Southern Expansion" strategy outlined in the Guangzhou Strategic Spatial Plan has been limited. It has not achieved the expected results in terms of facility construction, population attraction, industrial clustering, and value-added growth; (2) due to inherent limitations and a lack of planning support, the development of the new town presents a dilemma. Considering the current spatial structure, this paper reflects on the reasons for the failure of Guangzhou's strategic spatial planning, aiming to provide insights for the implementation of a new round of strategic spatial planning.

**Keywords:** strategic spatial planning; leapfrog; spillover; Guangzhou; Nansha





## 1. Introduction

Strategic spatial planning is a long-term framework aimed at addressing complex urban issues through the creation of a strategic vision and the implementation of short-term actions [1,2]. Against the backdrop of China's socioeconomic transformation, strategic spatial planning began to receive attention in China at the end of the 20th century [3]. Before the widespread popularity of strategic urban planning in China, Sir Peter Hall proposed megacities for the urbanization of the Pearl River Delta and Jakarta in East Asia [4]. Since around 2000, Guangzhou has also faced major opportunities for globalization development. During this time, urban areas faced urgent challenges related to major infrastructure development and regional coordinated growth. These challenges necessitated in-depth research and deliberation on critical issues such as urban spatial structure and expansion directions [5]. In this context, the procedural characteristics of master planning yielded unclear objectives and vague strategies, leading to a loss of utility [6]. Conversely, strategic spatial planning, with its flexibility and emphasis on strategy, is capable of swiftly and effectively providing solutions for urban development entities to address emerging issues [1,6,7]. It therefore gained widespread attention and recognition among local governments [5].

In 2000, Guangzhou took the pioneering step of formulating China's first strategic spatial plan, known as the "Guangzhou Strategic Spatial Plan". Subsequently, a multitude of outstanding strategic spatial plans emerged, significantly advancing urban planning in

China [8]. However, existing research has predominantly focused on the content of the "Guangzhou Strategic Plan" while lacking attention to its implementation effects [9–11]. To address this empirical gap, this study primarily explores the following questions: Is the current urban spatial pattern in Guangzhou consistent with what is mentioned in the "Guangzhou Strategic Plan"? What factors affect the effectiveness of strategic spatial planning implementation? Based on the research questions mentioned, this study utilizes a combination of theory and practice to analyze and identify the urban spatial pattern in Guangzhou, and summarize the factors influencing the results of strategic spatial planning implementation.

The structure of this article is as follows: The first section provides a brief introduction to the research background, research questions, and research methods. The second section reviews relevant research on the concept of strategic spatial planning and the evaluation of planning implementation. It also discusses debates related to leapfrog and spillover development. The third section introduces the research data and methods. The fourth part explores Guangzhou's urban spatial structure and the effectiveness of the 2000 Guangzhou Strategic Plan. The fifth part discusses the challenges presented by leapfrog development and elaborates on the dilemma confronting the Nansha new town. The sixth part summarizes this study and provides recommendations for future strategic spatial planning.

## 2. Literature Review

### 2.1. The Concept of Strategic Spatial Planning

According to existing scholarly viewpoints [1,12–17], strategic spatial planning is not a simple concept, process, or tool. It is a collection of concepts, processes, and tools developed based on specific circumstances, aiming to achieve ultimate goals. For example, Healey [13–15] has provided an in-depth analysis of the relationship between strategic spatial planning and local governance. She points out that the resurgence of strategic spatial planning is a response to the need for effective governance in the face of constantly changing political, economic, and social contexts. On the one hand, the evolving landscape demands that local governments implement "effective" governance in urban areas. On the other hand, within multi-level governance structures, there is a need to strengthen the voice of city governments and regional institutions. Consequently, strategic spatial planning becomes an avenue for government intervention in governance. Albrechts [1,2,16,17] defines strategic spatial planning as a social spatial process guided by the public sector. Through this process, a vision, behavioural approaches, and implementation methods are formulated, and through these means, the possibilities for the future of the region are delineated. In summary, strategic spatial planning serves as an action framework to guide the behaviour of different authorities, departments, and relevant private participants.

Some scholars state that the key difference between strategic spatial planning and a master plan lies in the fact that traditional spatial planning typically starts from controlling change, guiding growth, promoting development, and regulating development, ultimately resulting in land use planning [18–20]. Strategic spatial planning provides guidelines for overall development, leading to the formation of long-term visions and short-term actions that guide stakeholders in achieving common spatial transformation goals [1,2]. In terms of the content of planning, strategic spatial planning places emphasis on aspects such as spatial development direction, regional and urban development models, industrial selection, and functional positioning [6]. In terms of planning objectives, the positioning of strategic spatial planning is to grasp the broad trends and structures of future urban development through macro-level strategic spatial research [10,11]. In summary, compared to a master plan, strategic spatial planning places a greater emphasis on the depth and breadth of considerations related to a city's development structure, direction, and approach, allowing for a higher degree of freedom. And it places greater emphasis on integrating socioeconomic behavioural processes rather than solely focusing on land use, as traditional planning does [1,2,13].

### 2.2. Planning Implementation Evaluation

Currently, research on the evaluation of strategic spatial planning implementation can be broadly categorized into two main types: The first type is the evaluation of strategic spatial planning implementation results [21]. This category includes comprehensive evaluations as well as specialized assessments focusing on aspects such as demographics or industries [22,23]. It also encompasses comparative studies of the current status of urban strategic spatial planning and implementation [24]. The research data for such studies include static data, such as the distribution of points of interest (POI) [25], population distribution [26], and business locations [27], as well as dynamic interconnected data, including commuting patterns. For instance, Deng [25] identified spillover development patterns in Shenyang based on POI data. Xiao [26] utilized mobile phone signaling data to identify leapfrog development patterns in Shanghai.

The second type focuses on analyzing the factors that influence the effectiveness of strategic spatial planning implementation [28–32]. These studies take into account the various uncertainties inherent in the implementation process and employ methods such as surveys [28], literature reviews [15], and policy analyses [29–31] to examine the environmental, contextual, and procedural aspects of planning implementation. Alterman [32] pioneered a mixed-methods approach that categorizes factors influencing planning implementation into three groups: political and institutional factors, planning-related factors, and urban system-related factors. Sabatier [33] qualitatively identified a favourable economic, legal, and political environment as critical for ensuring successful planning implementation. Healey [13–15] conducted extensive literature reviews and interviews in a study involving 23 regions facing various planning challenges. By delving into process mechanisms to explain the outcomes, Healey collected valuable experiential information on planning implementation based on numerous diverse case studies.

In summary, existing research on planning implementation evaluation has shifted from a focus on singular "results assessment" to a rational emphasis on diverse "process monitoring".

### 2.3. Academic Debate: "Leapfrog" vs. "Spillover"

The identification and adaptation of spatial development patterns constitute a critical aspect of strategic spatial planning [13]. After centuries of development, cities have accumulated a wealth of spatial growth patterns. These patterns can primarily be categorized into two types [34]: "leapfrog" (Figure 1a), comprising models such as "New Town". The other type is "spillover" (Figure 1b), represented by models such as London's exemplary "Ring Road + Satellite Cities" and the "Radial Development" model typified by Copenhagen. However, it is essential to note that these patterns are not exhaustive, and some cities (like Randstad) may adopt hybrid or unique models [35].

#### 2.3.1. The Leapfrog Pattern

O'Sullivan [36] points out that as urban scale increases, urban challenges do not follow a linear growth trend but exhibit an accelerating pattern. Liu [37] said that, in this context, the "leapfrog" development pattern can effectively enhance urban economic performance and is currently a common trend in the development of large cities. Zhang [38] said that the Leapfrog pattern can help avoid the drawbacks of excessive population and industrial concentration, thereby enhancing the quality of the living environment. Furthermore, the leapfrog pattern can expand urban development space, increase urban land utilization efficiency, and promote a balance between work and residential areas in cities [39,40]. For example, Arribas-Bel [41] found that compared to "spillover" cities, "leapfrog" cities have larger areas, higher employment density, and higher per capita income levels by measuring employment centers in American cities. Zhang [38] measured the spatial performance of Chinese cities adopting the "leapfrog" pattern and found that leapfrog development is effective in improving urban spatial performance.

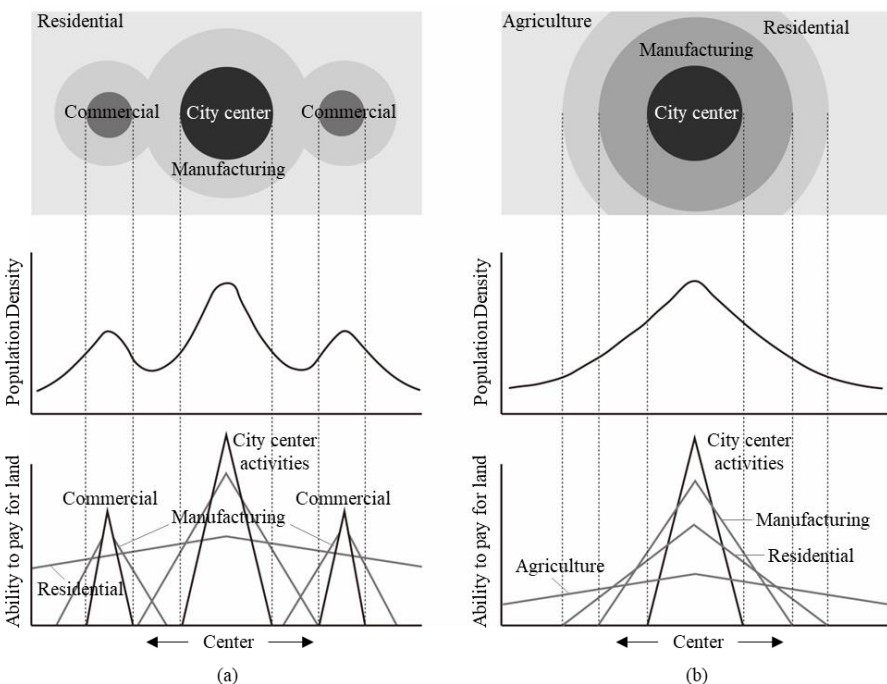

**Figure 1.** Two types of development models: (**a**) leapfrog and (**b**) spillover [34].

However, some scholars believe that leapfrog development also has its drawbacks [42–45]. These scholars propose that, while there are several research studies based on the leapfrog, including garden cities, new towns, and the theory of organic decentralization, that expound the benefits of leapfrog, these theories have not provided enough data nor the necessary scale for a more holistic examination [43–45]; thus, the superiority of the leapfrog to the spillover has not been definitively established. Parr's research [46], however, suggests that under the leapfrog pattern, benefits such as density, proximity, and informal communication, which are present in spillover structures of equivalent sizes, are diminished. Moreover, the leapfrog pattern may cause the relocation of traditional city center functions and lead to the decline of central areas [47]. For instance, the UK and European countries embarked upon numerous, large-scale new town construction projects. However, as urbanization reached a saturation point, the rapid momentum of urban development quickly dissipated, resulting in the cessation and even abandonment of many of these projects [47].

More importantly, in the leapfrog pattern, the distance between the new town and the city center is determined by the interaction of two conflicting forces [10,47]. The first force is the natural inclination, for economic and social reasons, to remain as close to the city center as possible. The second force is the planners' intention to provide physical and social distance from the city center. Abundant practical evidence demonstrates that even with a reasonable development direction, if the distance of the leap is unreasonable, the effects can be greatly compromised [48,49]. If the leap is too small, the city will continue to exhibit spillover development patterns [10]. If the leap is too great, it can surpass the city's own capacity, leading to an urban fiscal crisis [10,50], which means that the government invests significant funds in urban development but fails to achieve economic benefits, leading to financial instability within the government [51,52].

### 2.3.2. The Spillover Pattern

Some scholars argue that in certain situations, spillover development can be a relatively efficient spatial pattern [42–45]. Under spillover development, different regions are closely connected, and infrastructure is concentrated, which can effectively improve facility utilization rates and save municipal construction investments [46,47]. For example, Zhang [53] examined the relationship between the spatial structure and economic performance of 10 city clusters in China and found that spillover cities had higher economic

performance. Chen [54] conducted research on the spatial structure and economic efficiency of 20 city clusters in China and similarly concluded that the spillover development can increase economic efficiency.

However, numerous studies have pointed out that spillover development also has some drawbacks [55,56]. They argued that in the past, some cities (including Guangzhou) lacked the determination to relocate to the city center from the old town, resulting in numerous urban problems such as congestion, severe deterioration, and unequal distribution of facilities [8,10]. The spillover model in mega-cities has reached its limit, and implementing the leapfrog model would ameliorate these urban problems [57]. This dichotomy is comparable to the limited growth capacity of individual humans (e.g., height cannot exceed 3 m) and the theoretically unrestricted growth of the human population through reproduction.

This section discusses two distinct urban development patterns: "leapfrog" and "spillover". In certain scenarios, the "leapfrog" pattern is shown to enhance economic performance, improve the quality of the living environment, expand urban development space, and optimize land utilization [35–41]. However, it may lead to challenges such as the relocation of city center functions and fiscal crises [42–47]. On the other hand, the "spillover" pattern, in specific circumstances, is considered to boost economic efficiency by connecting different regions closely and concentrating infrastructure [45–47,53,54]. Nevertheless, it can result in issues like traffic congestion and unequal facility distribution [55–57]. In conclusion, both "leapfrog" and "spillover" urban development patterns have their advantages and limitations, and the choice between them should be made based on the specific context and urban requirements.

### 2.3.3. Spatial Development Patterns in the "Guangzhou Strategic Spatial Plan"

Since around 1980, the Pearl River Delta urban cluster, with Guangzhou at its core, has experienced rapid growth, garnering international attention [58]. However, by the year 2000, Guangzhou faced significant challenges [59]. On one hand, as foreign investments gained prominence, Hong Kong rapidly supplanted Guangzhou's role in the Pearl River Delta's regional economy, causing a substantial decline in Guangzhou's status [11]. In 1980, Guangzhou accounted for 62.8% of the industrial output in the Pearl River Delta, a figure that plummeted to 24.4% in 1990 and further declined to 18.1% by 1999. On the other hand, Guangzhou's urban area, with a mere 1443.6 km$^2$ in 2000, faced challenges in optimizing its spatial layout [57]. The city's 1996 urban master plan, in its quest for development space, resulted in a northward expansion that intensified the prevailing city-centric development pattern, exacerbating issues such as increased development intensity in the old city, worsened traffic congestion, and amplified urban problems [57–59].

Under these challenging circumstances, Guangzhou encountered a new opportunity for development [60]. This opportunity arose from a reorganization of administrative divisions, with Huadu and Panyu transitioning from county-level jurisdictions to district-level ones in the year 2000. This change expanded the urban area from its original 1443.6 km$^2$ to 3718.5 km$^2$, offering a fresh prospect for the spatial expansion of Guangzhou.

During this period, the prevailing urban master plan became obsolete, and Guangzhou needed to promptly adjust its development strategy to address the emerging opportunities and challenges [58–60]. Traditional urban master planning, characterized by lengthy preparation times and a lack of flexibility, was ill-suited to the immediate requirements of Guangzhou [11]. Consequently, the Guangzhou Strategic Spatial Plan was formulated [6]. In contrast to the traditional urban master plan, the "Guangzhou Strategic Spatial Plan" delved deeper into issues such as the causes of Guangzhou's urban development, the transformation of spatial development patterns, the scale of population growth, the selection of spatial structures, and the strategic implementation and management, particularly with regard to the spatial pattern selection during a phase of rapid urban development [11,58–60].

During the formulation process of the Guangzhou Strategic Plan at that time, different scholars had significant differences in opinions regarding which spatial development model to choose [11]. Some planners [60] thought that if Guangzhou aims to improve the quality

of its urban construction, it must recreate a city structure that matches its scale. Therefore, Guangzhou should achieve leapfrog development by constructing a new town in the south rather than continuing its expansion in the east.

However, some planners [59] emphasize the importance of eastward extension in Guangzhou, arguing that the city's conditions may not be ideal for the construction of a new town and that pushing for leapfrog development is premature. Based on this, Tongji University [11] suggested that it is not advisable for Guangzhou to embark on large-scale expansion to the south in the near term. Premature large-scale expansion is likely to devalue the capital of central city land resources and trigger a decline in the real estate market.

In 2000, Guangzhou promulgated the final "Guangzhou Strategic Plan" scheme. The final scheme proposed a spatial orientation, deciding on "southward expansion, northward optimization, eastward extension, and westward combination" (Figure 2). However, within the context of urban competition and ecological preservation, the eastward extension and southward expansion have become the primary foci; southward expansion serves as a directive for the city's "leapfrog" development, moving away from the city center, and eastward extension primarily signifies the "spillover" development expanding eastward from the city center [61]. This actually did not make a choice between 'spillover' and 'leapfrog'. While the new "2023 Guangzhou Strategic Plan" emphasizes southward expansion, eastward extension has also been introduced [29]. This mirrors Guangzhou's oscillation over the past two decades between the eastward extension and southward expansion, reflecting the ongoing debate over whether to vigorously develop the south area.

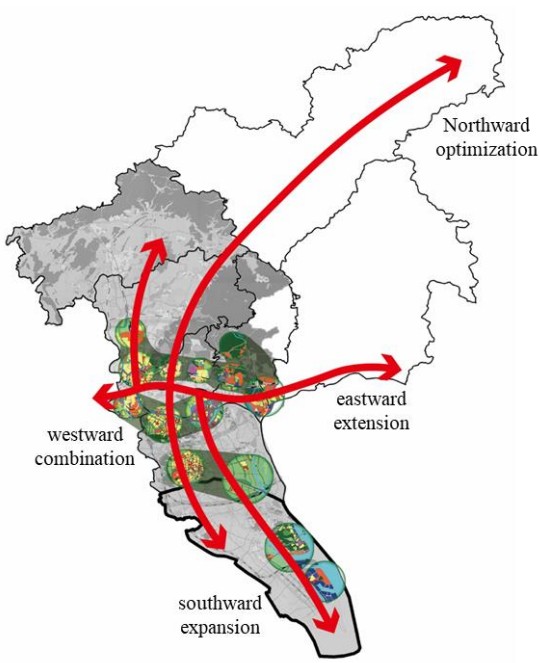

**Figure 2.** The urban development directions within the Guangzhou Strategic Spatial Plan in 2000.

## 3. Materials and Methods

### 3.1. Study Area

Guangzhou City is located in the south of China in the middle of Guangdong Province (Figure 3). It has 11 districts and covers an area of approximately 7434 km². Guangzhou City is the national central city and the center of commerce, culture, education, and transportation in South China [8]. In addition, it was also the first city in China to formulate a strategic spatial plan [6], making it of significant research importance.

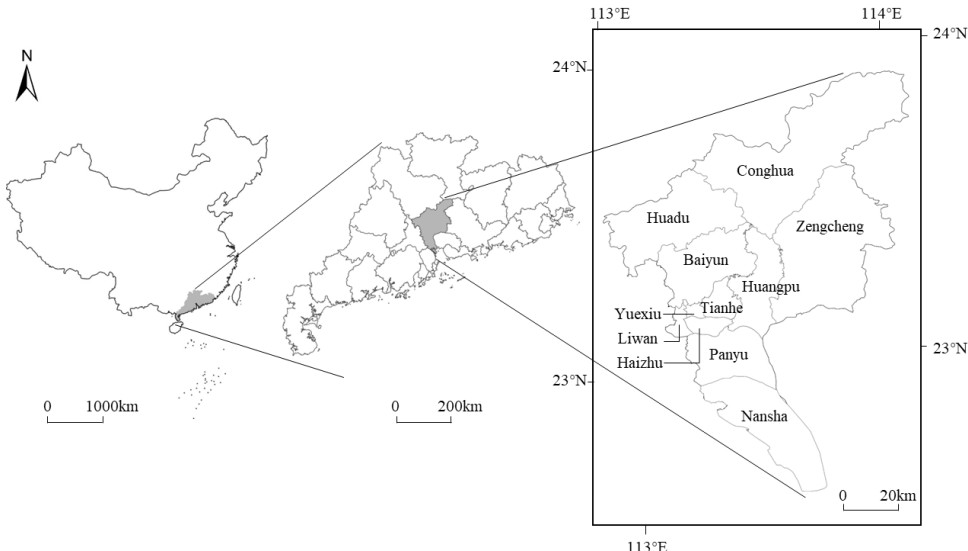

**Figure 3.** The location of the study area and its administrative divisions.

*3.2. Methods*

Based on the research question, and drawing on existing relevant studies [25–33], we approach the identification of urban spatial structures by examining three aspects: facility distribution, population distribution, and enterprise layout. We then compare these findings with the future spatial structure outlined in the "Guangzhou Strategic Spatial Plan". Subsequently, using policy analysis methods and incorporating the perspectives of relevant scholars and urban planners, we analyze the factors influencing the effectiveness of the strategic spatial planning implementation.

3.2.1. Distribution of Infrastructure

POI data offer advantages such as a large sample size and detailed information coverage [25]. The POI data used in this study were sourced from the Gaode open platform map (https://www.amap.com, accessed on 1 December 2021) and were collected in 2021. Each set of POI data includes information such as name, address, latitude, and longitude. Based on the city's different functions and in conjunction with the Gaode Map POI classification system, the POI data were categorized into the following four major classes: public services, catering/shopping services, recreational services, and financial/insurance services (Table 1). After geocoding, removal of duplicates, and spatial matching, a total of 375,123 POI data entries for the four categories were extracted, and their reliability was confirmed through sample on-site investigations.

**Table 1.** The types and quantity of POI data in Guangzhou.

| Category | Sub-Categories | Quantity | Percentage (%) |
|---|---|---|---|
| Public services | Healthcare services, government agencies, transportation facilities, educational and cultural facilities | 46,835 | 12.48% |
| Catering/shopping services | Dining services, shopping services | 297,470 | 79.29% |
| Recreational services | Sports and entertainment venues, vacation resorts and health retreats, cinemas | 24,858 | 6.63% |
| Financial/insurance services | ATMs, insurance, investment and financial management, banks | 5960 | 1.59% |

Kernel density estimation has found widespread application in the exploration of urban hotspots [62]. This method is used to calculate the density of spatial points or line

features within their surrounding neighborhoods, and it provides a continuous representation of the density distribution. This results in a kernel density value for each grid cell in an image, reflecting the distribution characteristics of spatial features. In this study, kernel density estimation is used to explore both enterprise data clusters and different types of POI data clusters in Guangzhou. The formula for calculating the kernel density function is as follows:

$$f(x) = \sum_{i=1}^{n} \frac{1}{\pi r^2} \varnothing (\frac{d_{ix}}{r})$$
(1)

In the formula: $f(x)$ represents the kernel density estimate at location $x$; $r$ is the search radius; $n$ is the total number of samples; $d_{ix}$ is the distance between POI point $i$ and $x$; $\varnothing$ is the weight for the distance.

### 3.2.2. Distribution of Residential and Workplace

Residence and employment are important factors in urban space, and the degree of their match greatly affects population distribution and land planning [26]. Cellular signaling data, a type of dynamic big data, primarily reflect changes in the dynamic connections of statistical objects [63]. They address issues such as the long sampling periods, high costs, and lags in updates associated with traditional methods.

This study utilized data from the telecommunication operator China Mobile, covering a period of 30 days from 1 June to 30 June 2019. The data were in anonymous form, with each set of signaling data containing user ID, timestamp, base station location code, event type (such as incoming and outgoing calls, incoming and outgoing text messages, and location updates), and other related information.

Mobile signaling data can be used to identify the residential and workplace locations of urban residents. The identification process is as follows: (1) Users who appear in Guangzhou for more than 60% of the days within a month are defined as permanent residents. The identification model is trained using historical nighttime mobile data from 20:00 to the next day at 8:00. Identify the area with the highest appearance probability that exceeds 60% as the user's residence. (2) Similarly, historical mobile data from working days (9:00 to 18:00) are used to train the identification model. Identify the area with the highest appearance probability that exceeds 60% as the user's workplace. Additionally, we consider expanding the sample size based on the market share of the mobile service provider.

### 3.2.3. Distribution of Enterprises

The spatial distribution of enterprises can reflect the industrial development of a city [27]. Enterprise data used in this study were sourced from the Qichacha Open Platform (https://www.qichacha.com, accessed on 1 December 2021) and were collected in 2021. The collected enterprise data include the name of each enterprise, its registered location, and the date of its establishment. After filtering, organizing, and compiling, a total of 1,747,284 enterprises of various types within Guangzhou were obtained. At the same time, within the research area, a 500 m × 500 m grid was established, and the number of businesses within each grid was counted to characterize the spatial distribution of enterprises. Additionally, the composition of gross domestic product (GDP) in various districts is sourced from the Statistical Yearbook published by the Guangzhou Municipal Bureau of Statistics in 2022.

### 3.2.4. Planning Documents Analysis

He [61] proposed that planning policies play a significant driving role in the development and evolution of urban spaces. General urban planning and strategic spatial planning documents serve as crucial foundations for guiding urban construction [30]. They represent the most direct means of expressing urban research and planning and serve as important documents for government officials, planners, developers, and the public to grasp the city's development context [29–31]. Therefore, we utilize a planning document analysis method and combine viewpoints from relevant planners and scholars to summarize the factors influencing the effectiveness of strategic spatial planning implementation, focusing

on two aspects: key industrial development and regional development focus. The policy documents mentioned in this article (Table 2) are sourced from government official websites, including the Guangzhou Municipal Planning and Natural Resources Bureau (http://ghzyj.gz.gov.cn/, accessed on 1 July 2023), The People's Government of Guangzhou Municipality (https://www.gz.gov.cn/, accessed on 1 July 2023), and the Department of Natural Resources of Guangdong Province (http://nr.gd.gov.cn/, accessed on 1 July 2023).

**Table 2.** The planning documents relevant to this study.

| Planning Type | Issuing Authority | Documents |
|---|---|---|
| Regional Development Planning | Central Committee of the Communist Party of China, State Council of the People's Republic of China, Guangdong Provincial Government | Outline Development Plan for the Guangdong-Hong Kong-Macao Greater Bay Area; Outline Plan for Reform and Development of the Pearl River Delta Region; Overall Plan for Deepening Comprehensive Cooperation with Hong Kong, Macao, and the Greater Bay Area in Nansha, Guangzhou |
| Urban Master Planning | Guangzhou Municipal People's Government, State Council of the People's Republic of China | Urban Master Plan of Guangzhou City, 1996–2010; Urban Master Plan of Guangzhou City, 2001–2010; Urban Master Plan of Guangzhou City, 2011–2020; Urban Master Plan of Guangzhou City, 2017–2035 |
| Strategic spatial planning | Guangzhou Municipal Planning and Natural Resources Bureau | Outline Plan for the Overall Strategic Concept of Urban Construction in Guangzhou; Guangzhou Urban Development Strategy Plan Towards 2049 |
| Nansha Regional Planning | | Overall Plan for Nansha Development Zone; Development Plan for Nansha Area; Development Plan for Nansha New Area, Guangzhou, 2012–2025 |

## 4. Results

### 4.1. Distribution of Infrastructure

The distribution of service facilities in Guangzhou presents an overall pattern of dispersion with local concentrations (Figure 4). High-density areas of these facilities are primarily located in the core districts around Tianhe, Yuexiu, and Haizhu. Beyond the core districts, the layout becomes more scattered, resulting in lower facility density. This distribution pattern is closely tied to the city's history. Traditional urban centers in Guangzhou, such as Yuexiu and Liwan, extended into core density areas that included Yuexiu, Liwan, Haizhu, and Tianhe. Subsequently, the city's developmental axis shifted eastward, especially with the construction of Tianhe, which drove the central urban core to expand toward the east. The Baiyun, Panyu, and Huangpu districts formed secondary density belts due to the ripple effect, while the Huadu, Conghua, Zengcheng, and Nansha districts developed as relatively dispersed peripheral centers. Nansha district exhibits a lower core density of facilities, reflecting the challenge of infrastructural radiation attributed to its considerable distance from the central area.

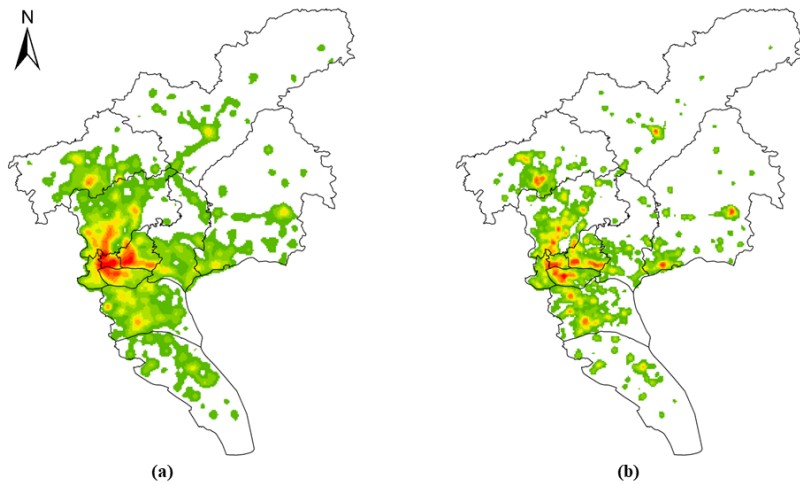

(a)　　　　　　　　　　(b)

**Figure 4.** *Cont.*

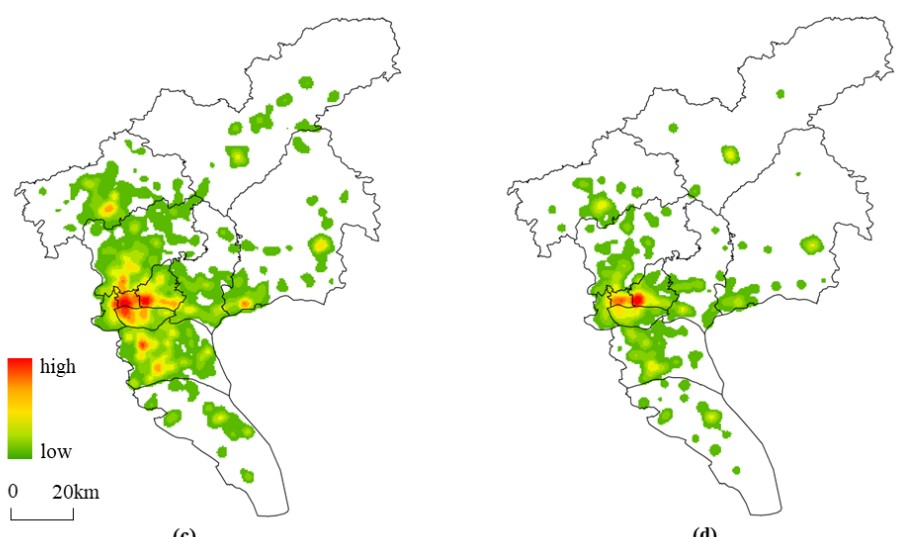

**Figure 4.** Density of four types of service facilities in Guangzhou: (**a**) public service facilities, (**b**) catering and shopping services facilities, (**c**) recreational facilities, and (**d**) financial facilities (the white areas in the figures represent 'no value').

### 4.2. Distribution of Population

Guangzhou's overall population density shows a fan-shaped expansion from the central urban areas towards the east and north (Figure 5). Although it has been traditionally regarded as two clusters in urban planning, in reality, it is mainly shaped by the constraints of mountains and rivers. Essentially, Guangzhou aligns with the spillover model. Furthermore, the outflow of the residential population exceeds that of the employment population, indicating that the development of the new center in Tianhe District has not fundamentally improved the urban structure. The decomposed functions of the old city center primarily consist of a large number of residential functions and a limited number of office functions. The substantial outflow of residential functions implies a greater reliance on the old city center, a lack of the development space needed for economic growth, and the inability of new functions to achieve sustainable and robust development.

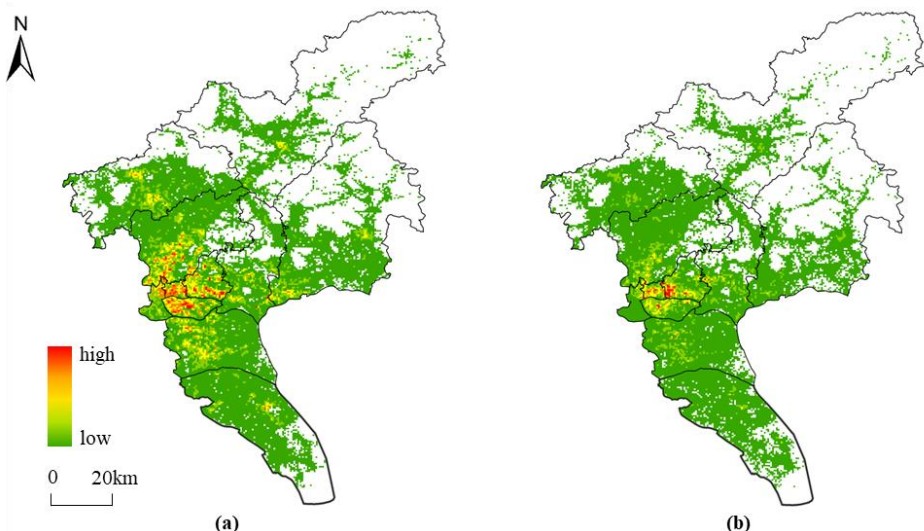

**Figure 5.** Population distribution of Guangzhou: (**a**) residential population and (**b**) employment population (the white areas in the figures represent 'no value').

The residential and employment populations of Nansha are significantly smaller than those of the city center (Figure 6a,b). They are relatively concentrated in the streets of

Nansha, and there are localized residential and employment hotspots in the area bordering Panyu District. Nansha District has fewer employment opportunities than it does residents, reflecting an insufficient supply of employment positions in Nansha.

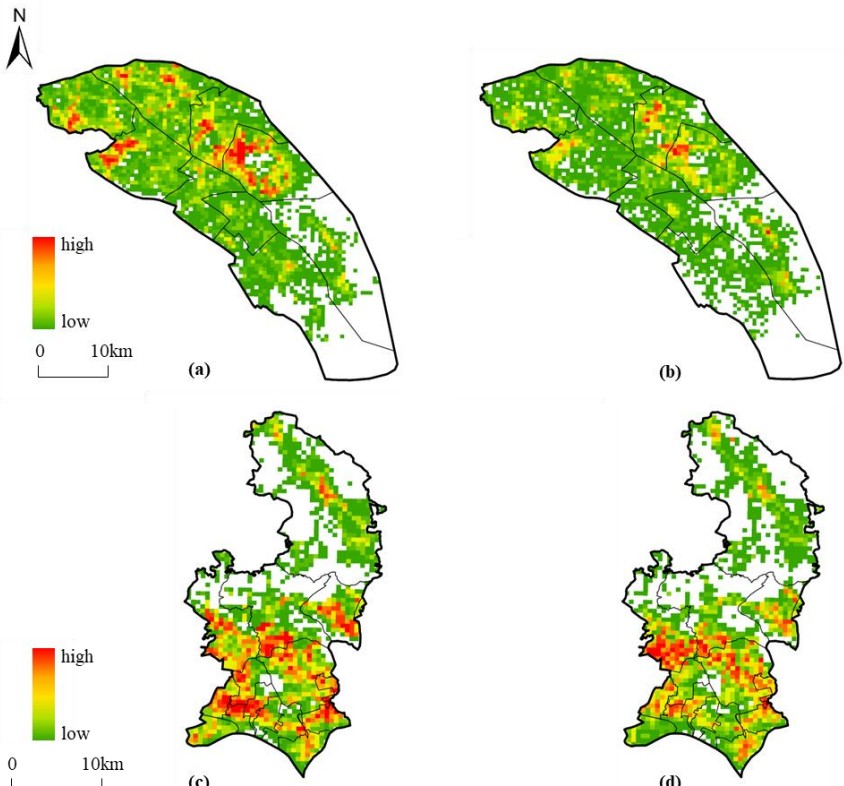

**Figure 6.** Population distribution of Nansha and Huangpu: (**a**) residential population of Nansha District; (**b**) employment population of Nansha District; (**c**) residential population of Huangpu District; (**d**) employment population of Huangpu District (the white areas in the figures represent 'no value').

Furthermore, in the Guangzhou Strategic Plan, the southern part of Nansha is designated for residential use and industrial zones, but it has relatively few residents and employees. This is mainly because Nansha did not adequately plan for office spaces, consumption areas, and residential areas in the early stages of development. As a result, many early-stage real estate projects and industrial zones have hindered the overall development and utilization of land in southern Nansha.

Huangpu District, especially at the southern border, where it adjoins Tianhe District, has a larger residential and employment population than Nansha District (Figure 6c,d). Guangzhou's eastward extension is thus progressing more rapidly than its southern expansion. This is primarily due to Huangpu District's strategic location as the geometric center of Guangzhou. It can effectively accommodate large-scale industrial migrations due to its ample land, absorb industrial overflow from the economic hub of Tianhe District, and further extend its influence eastward. Consequently, Huangpu District currently boasts higher population growth and inflow rates as well as rapid industrial development. Additionally, there is a discernible policy trend in Guangzhou that favors Huangpu District. Nansha District was designated as one of China's free trade zones in 2014, and this is widely recognized as one of Nansha's advantages. However, with the robust development momentum of Huangpu District, the Guangzhou municipal government is now actively pursuing the expansion of the free trade zone into Huangpu District, underscoring the pivotal role played by Huangpu District in the city of Guangzhou.

Urban operation and the daily commuting connections of residents can reflect the basic functional connections between the various districts of a city. The larger the commuting connections between different clusters, the higher the network connectivity, and the

stronger the functional connections. From the administrative district scale (Figure 7a), we see that Guangzhou has formed a multi-level, networked spatial structure with Tianhe District as the core and Baiyun, Panyu, and Haizhu as secondary centers. Among these, Tianhe District has the closest connections with Haizhu District and Panyu District, and Panyu District has become the pivotal point connecting the central urban area with Nansha District. Nansha New District is still developing gradually, and its position in the urban functional network is expected to rise incrementally.

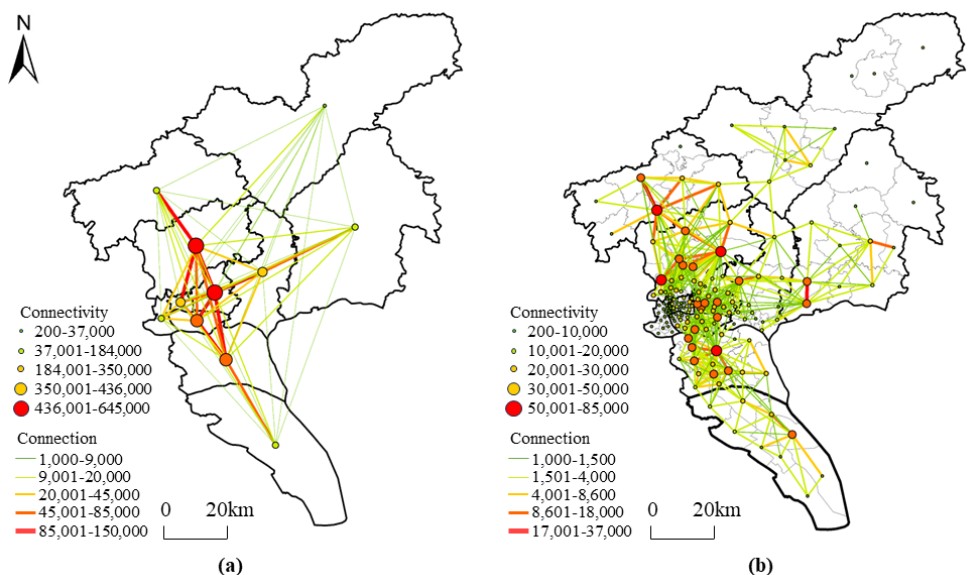

**Figure 7.** Commuting connections in Guangzhou: (**a**) inter-district commuting connections and (**b**) intra-district commuting connections.

At the street level (Figure 7b), the proportion of internal travel within each district is relatively high, and the central streets within a district often have the closest connections with other streets in the same district. Among these, Nansha Street demonstrates strong dynamic commuting connections with the rest of the streets in the district, reflecting the high functional interdependence of Nansha Street with its surrounding areas.

*4.3. Industrial Development*

Nansha District hosts a considerable number of enterprises, but the majority of them are small-scale enterprises in the wholesale and retail trade industry. In 2021, Guangzhou had a total of 1,747,284 enterprises, with Nansha District hosting 204,120 of them, accounting for 11.68% of the total and ranking third. In terms of the number of enterprises within Nansha District, the wholesale and retail trade industry has the highest number of enterprises (Figure 8), with a total of 107,277, making up 52.56% of all enterprises in Nansha. Following closely are the computer services industry and the leasing services industry, with 31,321 and 23,175 enterprises, respectively, accounting for 15.34% and 11.35% of all enterprises in Nansha. It is notable that industries highlighted for future development in the Guangzhou Strategic Plan, such as logistics, automobile manufacturing, and finance, have relatively few enterprises, making only 3.9%, 2.1%, and 6.8% of all enterprises in Nansha. This reflects a misalignment between actual industrial development and planned industrial focus.

While Nansha has a relatively large number of enterprises, the output values across various industries are not particularly prominent in the city. In 2021, Nansha District had a total output value of approximately CNY 160 billion, accounting for 5.6% of the city's total GDP and ranking seventh in the city. This reflects the fact that, while Nansha's favorable tax policies have attracted many businesses to settle there, they have not led to a significant increase in regional gross domestic product (GDP). As a free trade zone, Nansha offers a 15%

corporate tax incentive policy for certain industrial enterprises, attracting some relevant companies to establish a presence there. However, the industrial agglomeration effect in the new town is relatively weak, so many of these companies in Nansha are shell companies or registered offices. In reality, therefore, these Nansha enterprises do not generate output value. Taking financial enterprises as an example, China's financial industry relies heavily on policies and regional resources. Therefore, many financial enterprises in Nansha are newly established shell-registered financial institutions or branch offices that enjoy tax incentives. For instance, the Guangzhou Futures Exchange is registered in Nansha but operates from an office in Tianhe District. As a result, Nansha currently lacks a self-sustaining economic development path despite its considerable number of enterprises. By contrast, Huangpu District has achieved an annual average GDP growth rate of 7% for six consecutive years. With only 6.5% of the city's total area, it generates 40% of the city's industrial output value, 68% of the value of large-scale high-tech manufacturing, 15% of the GDP, and 21% of the tax revenue of the entire city (Figure 9).

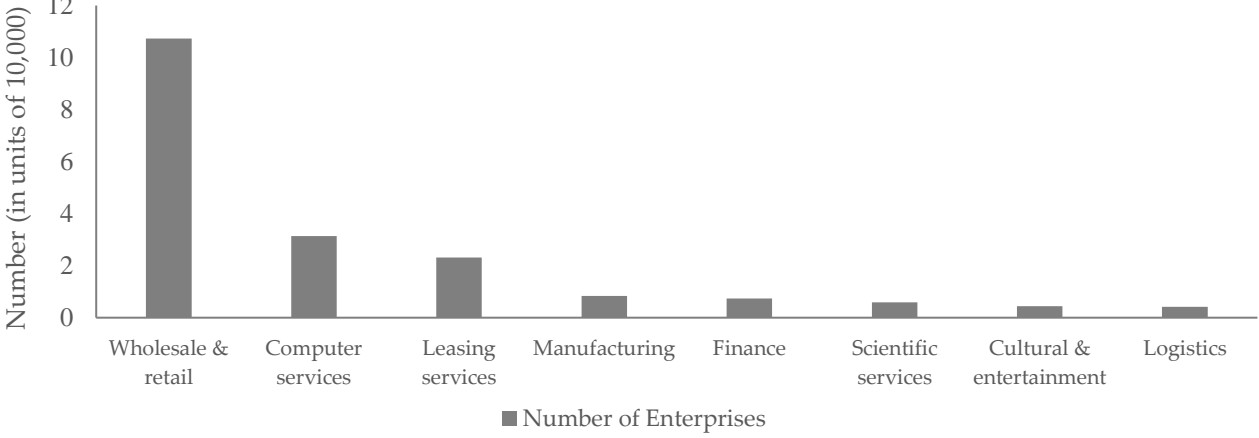

**Figure 8.** Number of enterprises in Nansha District.

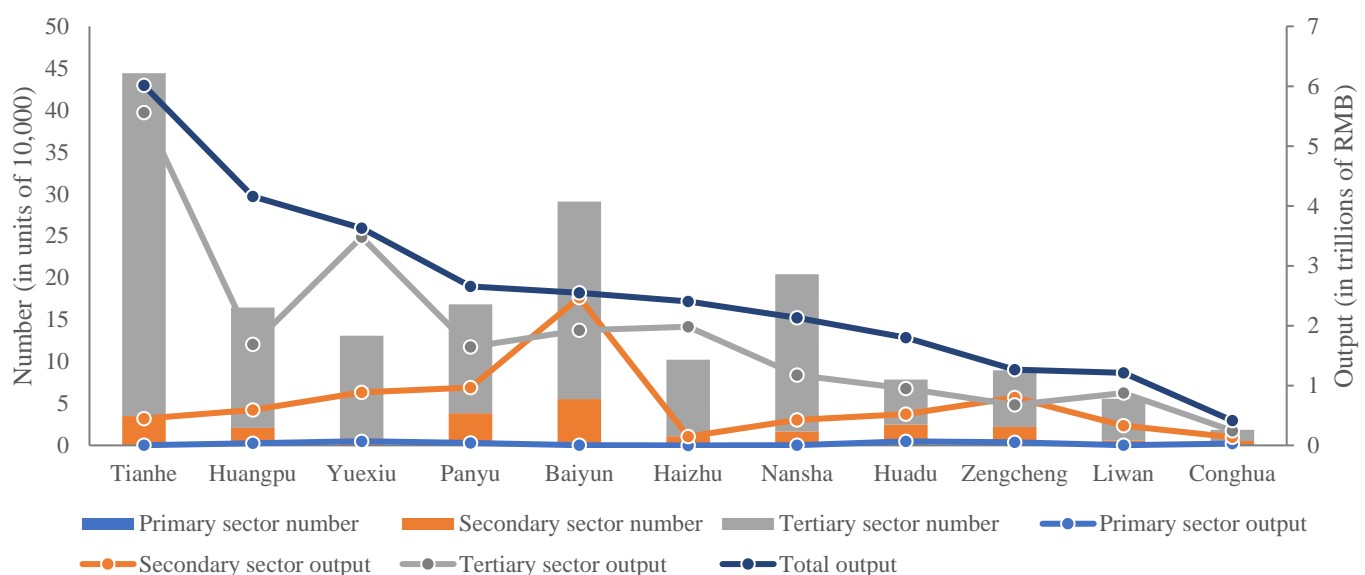

**Figure 9.** Number of enterprises and output value for the three industrial sectors in Guangzhou.

From the perspective of the overall spatial distribution of enterprises in the city, none of the three industries in Nansha District have formed industrial clusters with significantly higher agglomeration intensity than in other districts (Figure 10). Specifically, enterprises in the primary industry are distributed relatively evenly around the city. Although they

exhibit a tendency to form relatively concentrated cores in various district centers, the level of agglomeration is not very high. The spatial distribution characteristics of enterprises in the secondary and tertiary industries are similar, with companies highly concentrated in Tianhe District, Yuexiu District, and Haizhu District. Additionally, they extend along highways to the north, south, and east, forming small-scale agglomerations on the basis of axial distribution. Huangpu District, Baiyun District, and Panyu District serve as secondary agglomeration centers with relatively low agglomeration intensity. There is a noticeable gap between Nansha District and Panyu District, meaning that enterprises from the three industries have not expanded from the city center to Nansha District. From a regional perspective, Nansha District, Zengcheng District, and Conghua District are similar in that they are all in the outermost layer of enterprise distribution density.

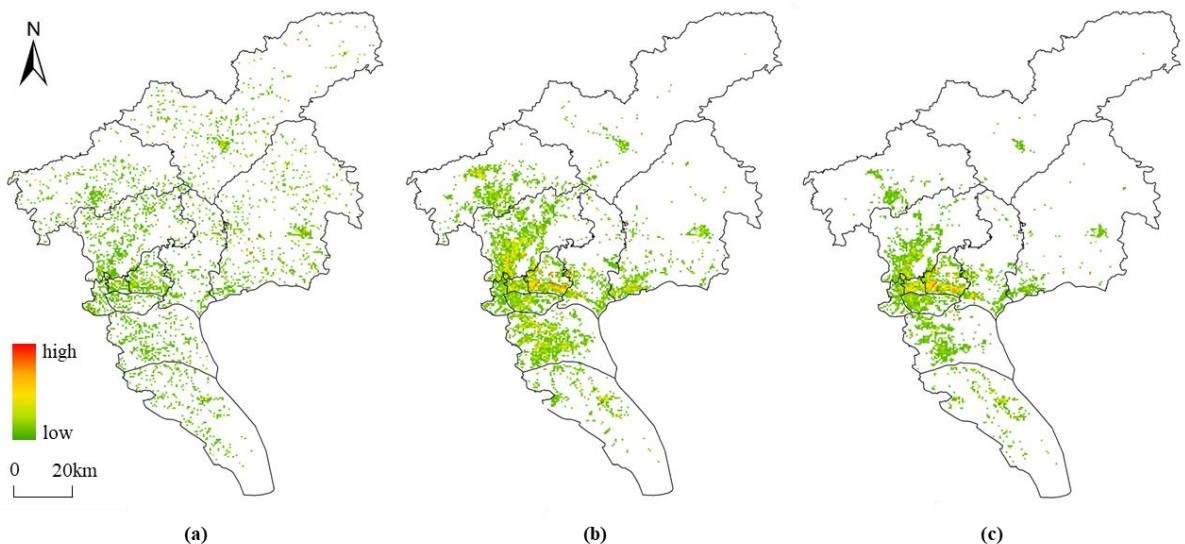

**Figure 10.** Distribution of three types of industries in Guangzhou: (**a**) primary industry, (**b**) secondary industry, (**c**) tertiary industry (the white areas in the figures represent 'no value').

The logistics industry, automobile manufacturing, finance, and high-tech industries are the key sectors for future development in Nansha. Examining the spatial distribution of these four industries, we observe that the core enterprises in Nansha's new town exhibit a concentrated but overall scattered pattern, forming an uneven spatial structure characterized by a single area with multiple points (Figure 11). This single-area aspect represents a high concentration of enterprises distributed in a spatially contiguous manner within Nansha Street, while the multiple-points aspect indicates scattered enterprise distribution throughout the entire area. Nansha Street is an area relatively concentrated on computer services, finance, logistics and manufacturing. As one of the earliest developed areas in Nansha, Nansha Street is currently the most mature sector, attracting a significant number of enterprises due to its well-established facilities.

Meanwhile, the northern part of Nansha, adjacent to Panyu District, has a larger number of enterprises, but their distribution is relatively scattered. The northern part of Nansha, although not planned as an industrial park in the overall plan, benefits from its spatial proximity to the central and northern parts of Nansha, which are closely connected to the Panyu urban area. Additionally, when Nansha was initially established, the central and northern area were integrated from Panyu District, and their infrastructure development, economic scale, population density, and various objective conditions were notably superior to other areas within the new district. This, to a certain extent, reflects the limited self-generating capacity of Nansha New Town, with its development primarily driven by external factors.

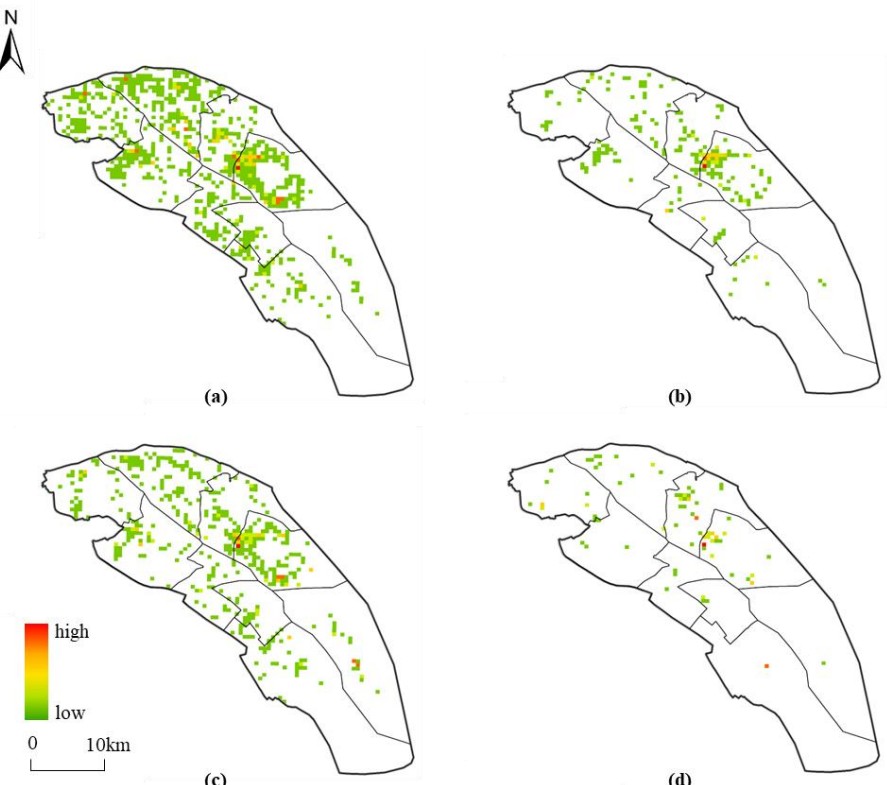

**Figure 11.** Distribution of four key industries in Nansha District: (**a**) computer services, (**b**) finance, (**c**) logistics, and (**d**) manufacturing (the white areas in the figures represent 'no value').

## 5. Discussion of Challenges

### 5.1. Why Did the Southward Expansion Fail?

Since the introduction of the "southward expansion "strategy in the 2000 Guangzhou Strategic Plan, after two decades of development, Nansha's infrastructure has remained relatively weak, its population vitality is low, and its industrial development is sluggish. In contrast, the spatial expansion of Guangzhou City has continued to spread spontaneously eastward (to Huangpu District) and northward (to Baiyun District), following a typical "spillover" pattern. Combining historical planning documents and the perspectives of relevant scholars and planners, we believe that the reasons for this situation can be attributed to three factors.

#### 5.1.1. Nansha's Inherent Limitations

Some studies [10,48,49] have shown that population density and distance from the city center are key factors for the success of new area development in "leapfrog" cities. This is because higher population density leads to closer connections between people and businesses, facilitating information sharing and exchange, and it also promotes the establishment of new enterprises [64–66]. Additionally, proximity to the city center signifies better market access, which is crucial for economic development [49].

According to the perspectives of some scholars [10,21,29,52,60,61,67], the distance of over 50 km between Nansha and the central area of Guangzhou has, to some extent, constrained Nansha's development. As revealed in Section 4, the results indicate that Nansha has significantly lower residential and employment populations compared to the central urban districts of Guangzhou (Tianhe, Haizhu, Liwan, Yuexiu) and the nearby central districts (Huangpu, Baiyun, Panyu). Meanwhile, the central urban districts have established close commuting connections with the nearby central districts, while Nansha exhibits weak commuting connections with other districts. Ye [67] found that the long commuting distance between Nansha and the central city is a significant deterrent for both

population and businesses to settle in Nansha based on a survey of Guangzhou residents. Han [52] points out that being farther away from the city center means farther away from the local market and higher commuting distances; such factors are not conducive to industrial development, population agglomeration, or the construction of public service facilities.

### 5.1.2. Multi-Level Government Orientation Differences

City governments often have a dual role as regional economic regulators and economic interest entities [68–70]. As regional economic regulators, city governments must fulfill various economic targets set by the central government [68]. As relatively independent economic interest entities, the city government is more inclined towards maximizing its own interests [69]. This can be seen from various policy documents issued by China (Table 3) [70]. Since 1993, Nansha has been designated as a national-level economic development zone in China. In 2023, the Chinese State Council released the Overall Plan for Deepening Guangdong-Hong Kong-Macao Greater Bay Area Cooperation in Nansha. However, for the Guangzhou government, while Nansha has a clear geographical advantage, its infrastructure, including transportation, needs improvement [60,61,67,69]. Large-scale projects outlined in the plan may take 5–10 years to complete, which could undermine local leaders' focus on achieving quick results [60,61,70]. By contrast, Huangpu District has a dense population and more developed infrastructure, allowing for rapid land allocation and increased fiscal revenue [60,61,71]. Therefore, in the interplay of these two forces, Huangpu and Nansha often coexist in historical Guangzhou master planning documents, serving as simultaneous focal points for urban development.

**Table 3.** Review of Guangzhou City planning documents since 2000.

| Year | Planning | Planning Focus |
|---|---|---|
| 2000 | Outline Plan for the Overall Strategic Concept of Urban Construction in Guangzhou | southward expansion, northward optimization, eastward extension, and westward combination; Achieving Urban Leapfrog Development |
| 2005 | Urban Master Plan of Guangzhou City, 2001–2010 | |
| 2016 | Urban Master Plan of Guangzhou City, 2011–2020 | Building a metropolitan area, two new city districts (Nansha and Huangpu), and three sub-centers (Huadu, Conghua, Zengcheng) |
| 2018 | Urban Master Plan of Guangzhou City, 2017–2035 | Setting Nansha as the sole sub-center of Guangzhou; Incorporating areas in Huangpu District (except for Jiulong Town) into the urban center |
| 2023 | Guangzhou Urban Development Strategy Plan Towards 2049 | Expanding to the south of the two seas (Nansha) and advancing to the east of the two rivers (Huangpu) |

### 5.1.3. Strategic Spatial Planning without Legal Constraints

Due to the lack of legal guarantees in Chinese strategic spatial planning, it is often the case that strategic spatial plans are easy to formulate but difficult or even impossible to implement and that they are prone to arbitrary changes [67,71]. This results in a situation in which, despite the flexibility of strategic spatial planning, there are few instances of coherent and consistent implementation [8].

This article has outlined the industrial positioning of Nansha in relevant planning documents since 1988 (Table 4). Industrial planning in Nansha has undergone several significant adjustments in response to external changes, such as the international financial crisis, regional transformation, and industrial upgrading in the Pearl River Delta. In 2000, influenced by the development strategy of southward expansion in Guangzhou, Nansha established its development direction: port strategy + industrialization + heavy industry. By 2009, considering factors such as ecological capacity and the transformation

of low-end manufacturing in the Pearl River Delta, Nansha abandoned the path of heavy industrialization. Then, in 2012, influenced by national policies, Nansha attempted to create an industrial cluster centered around eight major industries, including technological innovation, trade, financial services, and international shipping logistics. This shift in industrial positioning deviated significantly from the previous direction.

**Table 4.** Evolution of Nansha's industrial positioning.

| Year | Planning Documents | Industrial Positioning |
|---|---|---|
| 1988 | Conceptual Plan for Nansha | Logistics and freight transportation, tourism, science, commerce, and residential development |
| 1993 | Master Plan for Nansha Development Zone | Capital-intensive offshore processing |
| 2000 | Guangzhou Strategic Plan | Port + finance and high-tech industries + heavy industry |
| 2004 | Development Plan for Nansha (2005) | Ports and modern logistics + near-port industries, high-tech industries + steel, petrochemical, and automobile industries |
| 2012 | Development Plan for Nansha District (2012–2025) | Modern services, advanced manufacturing, high-tech industries |
| 2015 | Master Plan for Nansha (2015–2025) | Technology innovation industry, logistics, finance, international trade, high-end manufacturing |
| 2019 | Outline Development Plan for the Guangdong-Hong Kong-Macao Greater Bay Area | Technology innovation, financial services, international logistics |
| 2023 | Overall Plan for Deepening Guangdong-Hong Kong-Macao Greater Bay Area Cooperation in Nansha | |

In summary, because strategic spatial planning is not legally binding, it has resulted in a lack of clarity in Nansha's industrial development path [11,60,61,67]. Wang [29] has pointed out that the frequent changes in Nansha's industrial positioning in planning have led to issues such as weak connections between old and new industries, shorter industrial value chains, and lower levels of industrial clustering.

*5.2. The Southward Expansion Dilemma*

However, despite challenges such as its distance from the core city and weak development foundation, Nansha, as the Guangdong-Hong Kong-Macao Greater Bay Area's geometric center, bears the responsibility for promoting comprehensive cooperation. Therefore, the current development of Guangzhou is trapped in a dilemma: On one hand, Nansha can provide vast land for the development of Guangzhou, expanding the city's development space [6,11]. On the other hand, Huangpu is closely connected to the city center with a larger population and more businesses, holding potential for further development [61]. Additionally, Huangpu has concentrated infrastructure, which can effectively save municipal construction investments [58–60].

With limited financial resources, directing the majority of resources to Nansha for full-scale new town development presents several challenges [52]. It would require significant upfront costs and might not generate immediate benefits for the city. Simultaneously, the development of the main urban area could be postponed [11]. On the other hand, allowing the city to expand autonomously without focusing on Nansha could lead to disorderly

sprawl and challenges in urban planning and governance [38]. This unplanned growth might exacerbate urban issues and result in missed strategic opportunities for Nansha as a center for the Greater Bay Area [44].

## 6. Conclusions

This study uses Guangzhou as a representative case of urban strategic spatial planning in China and analyzes the implementation of Guangzhou's strategic spatial planning from the perspectives of facilities, population, and industries. It also utilizes policy documents and incorporates the viewpoints of relevant scholars and planners to explore the factors influencing the outcomes of strategic spatial planning implementation. The findings are as follows:

(1) The effectiveness of the "Southern Expansion" strategy outlined in the Guangzhou Strategic Plan has been limited. It has not achieved the expected results in terms of facility construction, population attraction, industrial clustering, and value-added growth. This indicates that there are challenges in the practical implementation of strategic spatial planning;

(2) The reasons for the ineffectiveness of the "Southern Expansion" strategy can be summarized into three categories: inherent limitations in Nansha's development, the lack of legal safeguards for strategic spatial planning, and differences in the value orientations of various levels of government. The city's spatial structure still exhibits a "spillover" pattern. In consideration of urban development trends and planning requirements, Guangzhou will still face a dilemma in the future.

This study illustrates that even strategic spatial planning, with its broad and flexible structure, can lead to contentious results under circumstances involving value conflicts and inherent limitations. However, regarding the issue of the strategic spatial planning's ineffectiveness in Guangzhou, we believe that the failure of certain strategies does not necessarily reflect the quality of strategic spatial planning practice, nor does it imply a return to the practices of master planning. What is more important is the reflection on the effectiveness of strategic spatial planning and the influencing factors. In other words, while Guangzhou's case may have its specific characteristics, it offers a crucial lesson: in the formulation and implementation of strategic spatial planning, it is essential to consider a comprehensive range of social, economic, and environmental factors to ensure the achievement of sustainable urban development outcomes. This also emphasizes the need for improvement and adaptation in strategic spatial planning practices to meet evolving conditions and carefully consider current urban development trends, and to, when appropriate, support these urban development trends rather than attempting to forcefully impose a new spatial framework. For instance, in the question of whether to construct new cities, one should take into account that there are many cases globally of failed new towns or new cities—and so new town/city development is a highly risky enterprise—to avoid potential issues and failures.

In summary, this study uses Guangzhou as an example and explores key issues in the implementation of strategic spatial planning. These issues encompass the inherent limitations of new districts, the challenges of multi-level government coordination, and the legal safeguards for strategic spatial planning. These findings emphasize the complex challenges that strategic spatial planning may encounter during practical implementation, offering a fresh perspective for urban strategic spatial planning theory. Furthermore, this study provides new insights into the implementation of strategic spatial planning and policy formulation, highlighting the importance of the need for strategic spatial planning flexibility. It also presents innovative development thinking that can assist governments in formulating more feasible planning strategies and contribute to achieving urban sustainable development goals.

**Author Contributions:** M.Z.: Conceptualization; methodology; formal analysis; visualization; writing. Y.Y.: writing; funding acquisition; resources. G.S.I.: writing; editing; formal analysis. All authors have read and agreed to the published version of the manuscript.

**Funding:** This research was supported by the National Office of Philosophy and Social Sciences (22&VHQ009).

**Data Availability Statement:** The data presented in this study are available on request from the corresponding author.

**Conflicts of Interest:** The authors declare no conflict of interest.

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
