# Peer review of "Reflection on Guangzhou’s Strategic Spatial Planning: Current Status, Conflicts, and Dilemmas"

_land, doi:10.3390/land12111996_

Round 1
Reviewer 1 Report
Comments and Suggestions for Authors
Dear authors,
This is an interesteing paper. However, I think it is badly organised in terms of consistency between methods and the data presented and discussed. I would suggest revising it or reshaping it to be split into two different papers. I've attached a file with more detailed observations.
Regards,

Author Response
Dear reviewer:
Thank you very much for your suggestions. We have made significant revisions based on your advice. Please refer to the attachment for specific details.
Best regards,

Reviewer 2 Report
Comments and Suggestions for Authors
Overall, this is a good paper. It is quite well written. Also, the empirical work is well presented and makes a convincing case for the eastward expansion of Guangzhou rather than leapfrogging into a new town in the south.
I do think however that the paper would benefit from an improved conceptual beginning and a strengthened conclusion.
The abstract and ln 32 makes the throwaway comment that strategic planning was initiated in England in the 1960s. This is questionable and the paper would be improved by a paragraph on strategic spatial planning, explaining its origins, and more clearly what it is and how it differs from master planning or comprehensive spatial planning. Note the term ‘strategic spatial planning’ which differentiates your subject from ‘strategic planning’ which is a practice used commonly in the corporate world and which emerged largely in the USA in the 1970s.
To assist with this, I suggest that you look at references such as:
· Albrechts, L. (2004). Strategic (spatial) planning reexamined. Environment and Planning B: Planning and design, 31(5), 743-758.
· Albrechts, L., Healey, P., & Kunzmann, K. R. (2003). Strategic spatial planning and regional governance in Europe. Journal of the American Planning Association, 69(2), 113-129.
· Healey, P. (2006). Relational complexity and the imaginative power of strategic spatial planning. European Planning Studies, 14(4), 525-546.
· Sartorio, F. S. (2005). Strategic spatial planning: A historical review of approaches, its recent revival, and an overview of the state of the art in Italy. DisP-The Planning Review, 41(162), 26-40.
· Albrechts, Louis. "Reframing strategic spatial planning by using a coproduction perspective." Planning theory 12, no. 1 (2013): 46-63.
· Albrechts, L. (2006). Shifts in strategic spatial planning? Some evidence from Europe and Australia. Environment and planning A, 38(6), 1149-1170.
· Todes, A. (2012). New directions in spatial planning? Linking strategic spatial planning and infrastructure development. Journal of Planning Education and Research, 32(4), 400-414.
I do think you need to indicate why the Guangzhou plan is a strategic spatial plan and not a master plan, and provide a bit more detail, if you can, on the background to the Guangzhou plan, especially why it was prepared as a strategic master plan and not a master plan.
I think you should relook at your conclusion. Apart from the rather specific issue of whether Guangzhou should go east or south, what does your paper say to a wider audience. For me, it suggests that planning should take actual trends very seriously and, where appropriate, support these trends rather than trying to impose a new spatial framing. There are many cases globally of failed new towns or new cities, and so new town/city development is a highly risky enterprise.
What does the paper say about strategic spatial planning. This is more difficult because a misplaced choice by the planners of Guangzhou does not necessarily reflect poorly on the practice of strategic spatial planning or suggest a return to master planning. However, it does suggest that even strategic master planning, despite its broad and flexible structure, can produce questionable outcomes and does require careful review.
Finally, please look again at Figure 2. Am I correct in concluding that a & b in the diagram are incorrectly placed and need to be swopped? For me, (a) represents the leapfrog and not (b)
Author Response
Dear reviewer,
Thank you very much for your suggestions. We have made significant revisions based on your advice. Please refer to the attachment for specific details.
Best regards,

Reviewer 3 Report
Comments and Suggestions for Authors
The article is very interesting and personally it was really enjoyable to read. I just have a few small annotations that can improve the quality of the article.
LINE 47: “Conversely, strategic planning, with its strong focus, flexibility, and emphasis on strategy, proved to be capable of swiftly and effectively providing tailored solutions for urban development entities to address emerging issues.” - This need references, not even only one but two or three references supporting this.
LINE 71: I know that this is being written on the first part, so the temptation is to start explaining the structure of the article going forward, but, if the authors want to keep this paragraph, I would suggest start by explaining what is being done in the first part also.
LINE 110: “Theories based on the leapfrog, including garden cities, new towns, and the theory of organic decentralization, are idealistic and have not been subjected to rigorous examination;” - Although that is up to a point fair to say, there is research (https://doi.org/10.3390/su14095029) that has already quantitatively analysed the evolution of Garden Cities to Social Cities - Garden Cities connected with each other, forming a polycentric and a leapfrog urban development. My suggestion is to sweeten the “have not been subjected to rigorous examination” and/or if say that research has been made into this area but does not provide enough data nor the necessary scale for a more holistic examination.
LINE 133: Please clarify what is meant by “urban fiscal bankruptcy”
LINE 134-138: Shouldn’t this be a citation?
LINE 385: “For urban development, the 30-kilometer mark is a critical distance.” - This needs to be referenced by more than one citation. The article did not reach this conclusion, thus this claim needs to be well supported by other research.
LINE 387: “Once a new town exceeds this distance, the population from the central city simply cannot commute to it.” - Again, this is a very strong affirmation, written as a fact without any references to supported. Either delete it or claim this affirmation by presenting more than one significant citation.
LINE 388: “This is the underlying logic that determines the success or failure of every new town” - At best case, this is ONE of the reasons that can provide context to wether a new town can be developed successfully or not. This has to be removed from the text.
Author Response
亲爱的审稿人,
非常感谢您的建议。我们根据您的建议进行了重大修改。具体详情请参阅附件。
此致,

Round 2
Reviewer 2 Report
Comments and Suggestions for Authors
Thank you for the positive response to my comments. They had been adequately addressed.
Author Response
Thank you.